# Investigation of the inter-rater reliability of three different plaque indices used in patients with fixed orthodontic appliances

Christina Erbe[1]*, Teresa Temming[2], Daniela Ohlendorf[3], Irene Schmidtmann[4], Priscila Ferrari-Peron[1], Ambili Mundethu[1], Heinrich Wehrbein[1]

**1** Department of Dentofacial Orthopedics & Orthodontics, University Medical Center of the Johannes Gutenberg-University, Mainz, Germany, **2** Department of Orthodontics, Clinic of Dentistry, Philipps University Marburg, Marburg, Germany, **3** Institute of Occupational, Social and Environmental Medicine, Goethe-University, Frankfurt, Germany, **4** Institute for Medical Biostatistics, Epidemiology and Informatics (IMBEI), University Medical Centre of the Johannes Gutenberg-University, Mainz, Germany

* erbe@uni-mainz.de

## Abstract

### Background/Objectives

To analyze the inter-rater reliability of three different plaque indices with regard to raters' orthodontic experience.

### Materials/Methods

The study analyzed 50 photographs of patients with maxillary and mandibular multibracket appliances (MB), captured via *Digital Plaque Imaging Analysis* (DPIA) for plaque assessment. Three indices - the modified Turesky index (TQH index), Attin index, and modified *bonded bracket index* (mBB index) were used. Fourteen evaluators with varying orthodontic experience levels (four with limited, five with moderate, and five with extensive experience) assessed the images.

### Results

The highest agreement among the evaluators in terms of ICC was obtained using the Attin index and the mBB index. The TQH index yielded the poorest agreement among evaluators. Orthodontic experience had no significant effect on inter-rater reliability. The evaluators with little orthodontic experience scored best in agreement with the Attin index, and the evaluators with much orthodontic experience scored best with the Attin and mBB indices. No difference was observed between the three plaque indices among the evaluators with moderate orthodontic experience.

### Limitations

Consistent classification of subjects into the same oral hygiene category by multiple raters using a plaque index was difficult. Consequently, calibration of raters in

**Data availability statement:** All data generated or analysed during this study are included in this published article.

**Funding:** The author(s) received no specific funding for this work.

**Competing interests:** The authors have declared that no competing interests exist.

practice may lead to a more unanimous classification of patients into the same oral hygiene category. A disadvantage of the classification of plaque values into oral hygiene categories was posed by the category boundaries. These resulted in two plaque indices being less consistent than was the case in reality.

## Conclusions/Implications

We recommend the use of an orthodontic plaque index (Attin index) or a combined dental and orthodontic plaque index (mBB index) in patients with an MB. The TQH index, considered the international standard plaque index, is less appropriate for application in these patients. Training and calibration of evaluators are of great importance when applying conventional plaque indices.

---

### Introduction

In Germany, the Fifth Oral Health Study (*Deutsche Mundgesundheitsstudie DMS V*) published in 2014 found a decrease in caries in the population. The prevalence of tooth loss is expected to decrease by 72% from 1997 to 2030 [1]. However, in patients treated with multibracket appliances (MB), an increase in plaque accumulation and gingivitis was observed [2,3]. The irregular surface of the MB appliance restricts the physiological cleaning function of oral muscles and saliva [4]. Furthermore, studies have shown that patients with MB appliances exhibit altered microbial flora [5–7]. Particular predilection sites for plaque accumulation include the areas behind arch wires [8], areas under bands of washed-out cementum [9], composite surfaces adjacent to the bracket base, the areas under the wings and in the slots of brackets, and composite-enamel interfaces [10]. Persistent plaque accumulation may result in the formation of demineralization, so-called *white spot lesions* [11], and the development of gingivitis [12].

Plaque indices serve as crucial tools for quantifying plaque accumulation and are widely employed in research to evaluate the effectiveness of oral hygiene products in various scientific studies [13–16]. In clinical practice, they also play a role in motivating patients to enhance their oral hygiene routines [17]. In patients with MB appliances, plaque tends to accumulate in different predilection sites compares to patients without MB appliances. Current studies focus on computer-assisted plaque assessment using planimetric quantification by plaque image analysis as an accurate, truly quantitative method [18,19]. However, these techniques are costly and necessitate complex equipment. In addition, a recent study showed conflicting results in accurately quantifying plaque in patients with MB [20]. In contrast, conventional plaque indices are quick and easy to use. However, few studies have specifically investigated the application of dental plaque indices in this population [21]. Furthermore, there is a scarcity of data on the reliability and reproducibility of orthodontic plaque indices in clinical research settings [21], and direct comparisons between standard dental plaque indices and those tailored for orthodontic patients remain infrequent [21].

The aim of this study was to investigate the inter-rater reliability of three different plaque indices used in orthodontics in patients with MB. In addition, the effect of the orthodontic experience of the raters on the results was analyzed. Each plaque index represented a plaque index system.

## Materials and methods

### Study design and subject matter

This study utilized a set of n = 50 photographs (accessed in June 2020 for research purposes, with no identifying patient information) captured using *Digital Plaque Imaging Analysis* (DPIA), a computer-assisted photographic analysis method which provides swift data collection and excellent reproducibility, as described by Klukowska et al [3]. The frontal and lateral intraoral photographs of the vestibular and lingual teeth surfaces were taken with a Nikon D80 reflex camera (Nikon Corporation, Tokyo, Japan), using cheek retractors and corresponding mirrors. Each photograph represented a distinct subject, resulting in a total of 50 subjects with a mean age of 13.9 ± 1.98 years. All subjects had previously been examined at the Department of Dentofacial Orthopedics and Orthodontics, University Medical Center of the Johannes Gutenberg-University, Mainz (Rhineland-Palatinate, Germany), and were fitted with a MB appliance in the both maxilla and mandible at the time of the study. Ethical approval was granted by the Freiburg Ethics Committee International (Baden-Württemberg, Germany) under feki code 07/2113. Detailed inclusion and exclusion criteria are available upon request from the Freiburg Ethics Committee. All subjects and their legal guardians provided verbal and written informed consent.

### Survey of plaque indices

In this study, the following plaque indices were examined as a representative of a plaque index system:

1) The TQH index is recognized as the international standard plaque index and is one of the most commonly used plaque indices in dental studies [22–25]. The Turesky index, along with the Quigley-Hein index, has also been frequently used in orthodontic studies [26–29]. The TQH index represents a modified dental plaque index. The classification of vestibular tooth surfaces of the TQH index were modified according to Cugini et al. [30] and divided into three regions (mesial, central, distal). Plaque accumulation was classified into the following assessment grades [24]:

- grade 0 - no plaque,
- grade 1 - single plaque islands along the gingival margin,
- grade 2 - thin plaque line (≤ 1 mm) along the gingival margin,
- grade 3 - plaque line > 1 mm, plaque covers ≤ 1/3 of the tooth surface,
- grade 4 - plaque covers ≤ 2/3 of the tooth surface,
- grade 5 - plaque covers > 2/3 of the tooth surface.

2) The Attin index was developed specifically for orthodontic patients with MB [31]. The development of the Attin index focused on the application of the plaque index in everyday clinical practice. This plaque index considers the predilection sites for plaque accumulation in patients with MB - the mesial, distal, and cervical areas of the bracket. Plaque accumulation was classified into the following assessment grades:

- grade 0 - no visible plaque,
- grade 1 - plaque islands on the proximal surfaces,

- grade 2 - in addition to the proximal surfaces, plaque islands cervical of the bracket,
- grade 3 - plaque covered > 1/3 of the surface cervical of the bracket.

3) The mBB index according to Delaurenti et al. [32] combines a dental and an orthodontic plaque index. The evaluation grades, adapted from the OHI according to Greene and Vermillion [33], represent the dental component and the four-part classification of the tooth surfaces (incisal, distal, mesial, gingival) according to Williams et al. [34] corresponds to the orthodontic component. Plaque accumulation was classified into the following assessment grades:

- grade 0 - no plaque,
- grade 1 - plaque covers ≤ 1/3 of the tooth surface,
- grade 2 - plaque covers ≤ 2/3 of the tooth surface,
- grade 3 - plaque covers > 2/3 of the tooth surface.

Plaque indices were collected by n = 14 evaluators using the DPIA photographs. The vestibular tooth surfaces of all twelve anterior teeth (six maxillary, six mandibular) were examined. The evaluators had varying levels of orthodontic experience. At the time of the study, n = 10 assistant dentists were undergoing further orthodontic training and n = 4 evaluators were orthodontic specialists at the Department of Orthodontics of the Johannes Gutenberg University Mainz.

The evaluators were divided into three groups according to

- little (≤ 1 year),
- moderate (≥ 2 years to ≤ 5 years), and
- extensive (> 5 years) orthodontic experience.

Accordingly,

- n = 4 raters had little orthodontic experience,
- n = 5 raters had moderate experience, and
- n = 5 raters had extensive orthodontic experience.

Each rater assessed the plaque accumulation of all n = 50 subjects and collected each of the three plaque indices once. All plaque indices were collected within a timeframe of two weeks, and the examinations were carried out at intervals of at least two days. All raters received a written description of the plaque indices in addition to a pictorial explanation for the evaluation.

## Statistical analysis

In this study, the results of plaque indices were summarized in Excel 2013 (Microsoft Office 2013, Microsoft Corporation, Redmond, WA, USA). R 4.4.1 (The R Foundation for Statistical Computing, 2016, download: https://cloud.r-project.org) was used in performing the statistical analysis. Conversion of all plaque values to percentage plaque values was achieved as follows:

$$\text{Plaque index (\% )} = \frac{\frac{\text{Total of plaque values}}{\text{Number of surfaces}}}{\text{Highest graduation score}} \times 100$$

The inter-rater reliability related to the classification of the subjects into oral hygiene categories was determined with Fleiss' Kappa for multiple raters [35] and interpreted according to Landis and Koch [36]. For this analysis, the percentage

plaque values of the four plaque indices were classified into the oral hygiene categories according to Lange [37]. Inter-rater reliability using the percentage plaque values was analyzed using the intra-class correlation coefficient (ICC) [38]. In this study, the ICC (2.1) was applied and interpreted according to Cicchetti [39]. The ICC was determined with and without consideration of the orthodontic experience of the raters.

## Results

Fleiss' Kappa showed moderate agreement among the raters for all three plaque indices according to Landis and Koch with regard to the classification of the subjects in the same oral hygiene category (Table 1). The evaluators rated the oral hygiene of the subjects with the TQH and Attin indexes somewhat more consistently than with the mBB index.

The raters achieved excellent agreement with the Attin index in relation to the congruence of the plaque values according to the Cicchetti interpretation. With the TQH and mBB index, there was good agreement among the evaluators (Table 2).

In this study, the Attin index was found to be most appropriate for the raters with little orthodontic experience. For the raters with moderate orthodontic experience, there was no significant difference between the plaque indices. For the raters with a lot of orthodontic experience, the Attin index and the mBB index were found to be the most appropriate (Table 3).

Table 1. Evaluation of the consistency of the oral hygiene categories among raters using Fleiss' Kappa with 95% confidence intervals.

| Plaque index | Fleiss' Kappa | 95% CI |
|---|---|---|
| TQH | 0.44 | [0.42; 0.47] |
| Attin | 0.44 | [0.41;0.46] |
| mBB | 0.38 | [0.36;0.41] |

Table 2. Evaluation of the consistency of plaque values among raters without considerations of their orthodontic experience with the ICC.

| Plaque index | ICC | 95% CI |
|---|---|---|
| TQH | 0.66 | [0.50; 0.79] |
| Attin | 0.75 | [0.64; 0.83] |
| mBB | 0.74 | [0.60; 0.84] |

Table 3. Evaluation of the consistency of plaque values among raters with consideration of their orthodontic experience with the ICC.

| Plaque index | Orthodontic experience | ICC | 95% CI |
|---|---|---|---|
| TQH | little | 0.63 | [0.29; 0.81] |
| | moderate | 0.76 | [0.59; 0.87] |
| | a lot | 0.68 | [0.34; 0.84] |
| Attin | little | 0.81 | [0.68; 0.89] |
| | moderate | 0.74 | [0.58; 0.84] |
| | a lot | 0.78 | [0.67; 0.86] |
| mBB | little | 0.76 | [0.52; 0.88] |
| | moderate | 0.77 | [0.55; 0.88] |
| | a lot | 0.76 | [0.43; 0.89] |

## Discussion

The use of plaque indices has been investigated less frequently in orthodontic studies than in dental studies. In 2010, Raggio et al. [40] analyzed the Silness and Löe index and the Turesky index concerning reliability and discriminatory ability in a dental study. Furthermore, Eaton et al. [41] and Kingman et al. [42] investigated the reliability of the Silness and Löe index. Quirynen et al. [43] tested the discriminatory ability of the Quigley-Hein index and other plaque indices in 1991. Matthijs et al. [44] specified the intra-rater reproducibility of the Turesky index and the *Navy plaque index* according to Elliott et al. [45] in 2001. Marks et al. [46] also analyzed the Turesky index for reliability and reproducibility in 1993. Few recent studies have examined the reliability of conventional plaque indices [47–50]. Paschos et al. [21] are among the few authors who investigated the reliability of plaque indices in subjects with MB.

The results of our study show that the consistent classification of subjects into the same oral hygiene category by multiple raters using a plaque index proved difficult. The classification of plaque into subdivided oral hygiene categories is of particular clinical importance. Oral hygiene categories provide an indication of a patient's oral hygiene status and help classify patients into an appropriate prophylaxis program. In daily clinical practice, the classification of patients into a prophylaxis program to improve or maintain oral hygiene is more important than a precisely determined plaque score. The calibration of multiple raters plays a minor role in the collection of plaque indices in practice as opposed to studies [51]. However, the present study showed that the subjects were classified into the same oral hygiene category by the raters with insufficient agreement. Consequently, the calibration of raters in practice may lead to a more consistent classification of patients into the same oral hygiene category. The correct classification of patients into the appropriate oral hygiene category is relevant in clinical practice during orthodontic treatment with MB, as it helps to maintain or improve oral hygiene in the best possible way. A disadvantage of the classification of plaque values into oral hygiene categories was posed by the category boundaries. These resulted in two plaque indices being less consistent than was the case in reality.

The inter-rater reliability of the TQH index compared to the mBB index was better than the subsequent analysis with the ICC. Due to the higher number of scoring levels of the TQH index, the probability of agreement between the subjects' oral hygiene categories was higher with the TQH index than with the mBB index. This may be the reason why, compared to the mBB index, the TQH index obtained a better result in the analysis of inter-rater reliability with Fleiss' Kappa than in the analysis with the ICC.

The results of the ICC showed that an orthodontic plaque index (Attin index) and a combined dental and orthodontic plaque index (mBB index) were more appropriate for the raters than a modified TQH index. Compared with the three-part vertical division of tooth surfaces of the TQH index, the four-part division of the mBB index possibly allowed a better assessment of plaque accumulation. In addition, the results indicated that the mBB index was easier to collect with a lower number of assessment grades than the TQH index. However, this meant that the mBB index lost precision. It should be noted that in this study the ICC was diminished by low variability in the subject population due to the inclusion criteria [52]. Orthodontic experience showed no significant effect on inter-rater reliability. The results of our study suggest that the number of dental surfaces to be evaluated may have played a role. The Attin index, which assessed the entire tooth surface, was apparently easier to collect for raters with little orthodontic experience than the mBB and TQH indexes, which assessed multiple tooth surfaces.

In 2012, Hefti and Preshaw [53] pointed out the importance of training and calibration of raters when collecting a plaque index in clinical trials. The authors also mentioned appropriate procedures in this regard [53]. In this study, the inter-rater reliability of plaque indices was evaluated without training and/or calibrating the raters. It can be assumed that training and calibrating of the raters would lead to better inter-rater reliability. Consequently, training and calibration are of great importance when collecting conventional plaque indices in studies. Before the start of a planned study, it must always be ensured that the raters are trained and calibrated.

Numerous studies on the reliability of plaque indices in dentistry were found in the literature [49,50,54]. However, there are only a few studies on the reliability of plaque indices in orthodontics. In 2014, Paschos et al. [21] analyzed the

use of four plaque indices in subjects with MB. The study examined the orthodontic Attin index, the *Modified Orthodontic Plaque Index* according to Paschos et al., the Quigley-Hein index, and the *Navy Plaque Index* according to Clemmer and Barbano [55]. The Quigley-Hein index represented a dental plaque index, as did the TQH index in this study. The *Modified Orthodontic Plaque Index,* similar to the mBB index in this study, consisted of a combination of a dental and an orthodontic plaque index. The results of Paschos et al. were comparable to the results of our study. The study by Paschos et al. also showed better reliability for the Attin index and the *Modified Orthodontic Plaque Index* than for the Quigley-Hein index. The study by Paschos et al. further examined the plaque indices in terms of orthodontic experience. Their study found that the Quigley-Hein index as a dental plaque index was more sensitively related to educational level than an orthodontic or combined dental and orthodontic plaque index. The results of our study and the study by Paschos et al. were consistent with the literature in that dental plaque indices are inappropriate for patients with MB. In 2001, Matthijs et al. [44] also pointed out difficulties in collecting the Turesky index. The aforementioned authors investigated the intra-rater reliability of the Turesky index in dental subjects. The study discussed the assumption that the raters changed the criteria for collecting the plaque index between two measurements. Matthijs et al. [44] mentioned the scoring levels of the Turesky index as a reason. The inter-rater reliability reported here is consistent with previous paediatric research, with an ICC of 0.68–0.88 and 0.48–0.77 for n = 2 raters [56]. In a study by Marks et al. [46], the Turesky index collected from n = 11 raters yielded an ICC of 0.70 [46]. The reason for the slightly better correlation compared with our study may have been the extensive training and calibration procedures of the raters. In addition, both studies were conducted on dental subjects.

In orthodontic studies, the Silness and Löe index is the most commonly used plaque index [57]. Like other plaque indices for orthodontic patients, this index was modified by Williams et al. [34]. Unlike all other conventional plaque indices, the Silness and Löe index assesses the thickness of the plaque. Silness and Löe [58] stated that the method of choice for distinguishing grade 1 and 2 of the Silness and Löe index is the use of a probe. Consequently, the collection of the Silness and Löe index with photographs is limited. In addition, the collection of this plaque index by multiple raters is complicated because the use of a probe results in damage to the plaque film [41]. For these reasons, the mBB index is a good alternative to the Silness and Löe index for scientific studies. In daily clinical practice, the focus is on the quick, easy application of plaque indices. Especially in patients with MB, the evaluation of oral hygiene and the motivation of the patient are of outstanding importance. Based on the results of this study, we recommend the use of the Attin Index, especially for everyday practice. The Attin index was developed specifically for daily use in the clinic and not for epidemiological and experimental studies [31]. In our study, the Attin index achieved comparable results to the mBB index in the evaluation of inter-rater reliability. In contrast to the mBB index, a plaque value per tooth is collected. This simplifies the practical application. The Attin index is thus well suited for orthodontic practice in patients with MB.

## Conclusion

The mBB index used as a combined dental and orthodontic plaque index, and the Attin index used as an orthodontic plaque index, achieved good agreement among the raters. Consequently, we recommend these two plaque indices for determining plaque accumulation in patients with MB. The TQH index, the international standard plaque index, achieved the poorest agreement among raters and is therefore not suitable for use in patients with MB. In principle, when using conventional plaque indices, special attention should be paid to the training and calibration of the evaluators in advance.

## Acknowledgments

We acknowledge Dr. Judith Hoffrichter, Highline PR & Publishing, Sarah Hanagarth, Johanna Thomé, Annalies von Redwitz and Juliane Schertler for their language editing services.

## Author contributions

**Conceptualization:** Teresa Temming, Daniela Ohlendorf, Irene Schmidtmann, Heinrich Wehrbein.

**Data curation:** Teresa Temming.

**Investigation:** Teresa Temming.

**Methodology:** Christina Erbe, Teresa Temming, Daniela Ohlendorf.

**Project administration:** Christina Erbe, Teresa Temming, Irene Schmidtmann, Ambili Mundethu, Priscila Ferrari-Peron.

**Resources:** Irene Schmidtmann.

**Software:** Daniela Ohlendorf, Irene Schmidtmann.

**Supervision:** Christina Erbe, Daniela Ohlendorf, Irene Schmidtmann, Ambili Mundethu, Heinrich Wehrbein, Priscila Ferrari-Peron.

**Validation:** Irene Schmidtmann.

**Visualization:** Irene Schmidtmann.

**Writing – original draft:** Christina Erbe, Teresa Temming, Heinrich Wehrbein.

**Writing – review & editing:** Christina Erbe, Teresa Temming, Daniela Ohlendorf, Irene Schmidtmann, Ambili Mundethu, Heinrich Wehrbein, Priscila Ferrari-Peron.

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
