## [Decision Letter · Decision Letter 0]

22 Oct 2024

PONE-D-24-15938Investigation of the inter-rater reliability of three different plaque indices used in patients with fixed orthodontic appliancesPLOS ONE

Dear Dr. Erbe,

Thank you for submitting your manuscript to PLOS ONE. After careful consideration, we feel that it has merit but does not fully meet PLOS ONE’s publication criteria as it currently stands. Therefore, we invite you to submit a revised version of the manuscript that addresses the points raised during the review process.

We look forward to receiving your revised manuscript.

Kind regards,

Sameh Attia, MS

Academic Editor

PLOS ONE

Journal Requirements: When submitting your revision, we need you to address these additional requirements. 1. Please ensure that your manuscript meets PLOS ONE's style requirements, including those for file naming. The PLOS ONE style templates can be found at https://journals.plos.org/plosone/s/file?id=wjVg/PLOSOne_formatting_sample_main_body.pdf and https://journals.plos.org/plosone/s/file?id=ba62/PLOSOne_formatting_sample_title_authors_affiliations.pdf 2. We noticed you have some minor occurrence of overlapping text with the following previous publication(s), which needs to be addressed: https://openscience.ub.uni-mainz.de/bitstream/20.500.12030/8919/1/comparison_of_different_plaqu-20230307140914730.pdf In your revision ensure you cite all your sources (including your own works), and quote or rephrase any duplicated text outside the methods section. Further consideration is dependent on these concerns being addressed. 3. We note that your Data Availability Statement is currently as follows: All data generated or analysed during this study are included in this published article (and its Supplementary Information files). Please confirm at this time whether or not your submission contains all raw data required to replicate the results of your study. Authors must share the “minimal data set” for their submission. PLOS defines the minimal data set to consist of the data required to replicate all study findings reported in the article, as well as related metadata and methods (https://journals.plos.org/plosone/s/data-availability#loc-minimal-data-set-definition). For example, authors should submit the following data: - The values behind the means, standard deviations and other measures reported;- The values used to build graphs;- The points extracted from images for analysis. Authors do not need to submit their entire data set if only a portion of the data was used in the reported study. If your submission does not contain these data, please either upload them as Supporting Information files or deposit them to a stable, public repository and provide us with the relevant URLs, DOIs, or accession numbers. For a list of recommended repositories, please see https://journals.plos.org/plosone/s/recommended-repositories. If there are ethical or legal restrictions on sharing a de-identified data set, please explain them in detail (e.g., data contain potentially sensitive information, data are owned by a third-party organization, etc.) and who has imposed them (e.g., an ethics committee). Please also provide contact information for a data access committee, ethics committee, or other institutional body to which data requests may be sent. If data are owned by a third party, please indicate how others may request data access. 4. We note you have included a table to which you do not refer in the text of your manuscript. Please ensure that you refer to Table 3 in your text; if accepted, production will need this reference to link the reader to the Table.

Reviewers' comments:

Reviewer's Responses to Questions

**Comments to the Author**

1. Is the manuscript technically sound, and do the data support the conclusions?

Reviewer #1: Partly

Reviewer #2: Yes

2. Has the statistical analysis been performed appropriately and rigorously?

Reviewer #1: Yes

Reviewer #2: Yes

3. Have the authors made all data underlying the findings in their manuscript fully available?

Reviewer #1: Yes

Reviewer #2: Yes

4. Is the manuscript presented in an intelligible fashion and written in standard English?

Reviewer #1: Yes

Reviewer #2: Yes

5. Review Comments to the Author

Reviewer #1: Dear Authors

the topic of the manuscript is interesting but some changes are necessary before taking it into consideration for publication. Here are my suggestions to imrpve it:

- try to better describe the DPIA procedure to ensure the anonymity of patients. Explain what the acronym means and briefly what it consists of ;

- better describe about the analyzed 50 photographs. Please explain whether the photo is of a single tooth or a group of teeth. Which surface (mesial, vestibular, occlusal...), what type of photo (intra or extra-oral, occlusal, lateral, 3/4,). What camera was used? Did you use flash? Who took the photos? Where the patients have been enrolled?

- please explain wh etegories: low, medium, and high orthodontic experience. how these 3 groups have been created?

- Who the observers are: hygienists, students, dentists, orthodantists, dental assistents?

- The observers were divided into 3 categories: low, medium, and high orthodontic experience. But how these 3 groups were created?

- Statistics is well done.

- references: 47/59 are older than 10 years old.

Please address these concerns.

Best regards

Reviewer #2: The authors find that the inter-rater reliability of three plaque indices varies based on the evaluators' orthodontic experience. The Attin and mBB indices show the highest agreement among evaluators, while the TQH index has the lowest. Orthodontic experience does not significantly affect reliability. Evaluators with little experience show the highest agreement with the Attin index. Calibration of raters is recommended to achieve more consistent classifications. The authors conclude that the Attin and mBB indices are preferable for patients with multibracket appliances, whereas the TQH index is less suitable. Training and calibration are essential for effective application of plaque indices.

I would like to the authors for their valuable efforts on this study. It was a pleasure to read.

6. PLOS authors have the option to publish the peer review history of their article (what does this mean? ). If published, this will include your full peer review and any attached files.

**Do you want your identity to be public for this peer review?** For information about this choice, including consent withdrawal, please see our Privacy Policy .

Reviewer #1: **Yes: ** Cinzia Maspero

Reviewer #2: No

---

## [Author Response · Author response to Decision Letter 1]

28 Jan 2025

Journal Requirements:

Concern of the reviewer: Please ensure that your manuscript meets PLOS ONE's style requirements, including those for file naming. The PLOS ONE style templates can be found at

and

Our answer: Thank you very much for your comments, we have implicated the changes suggested.

Concern of the reviewer: We noticed you have some minor occurrence of overlapping text with the following previous publication(s), which needs to be addressed:

https://openscience.ub.uni-mainz.de/bitstream/20.500.12030/8919/1/comparison_of_different_plaqu-20230307140914730.pdf

In your revision ensure you cite all your sources (including your own works), and quote or rephrase any duplicated text outside the methods section. Further consideration is dependent on these concerns being addressed.

Our answer: Thank you very much for your recommendation, we rephrased any duplicated text.

Concern of the reviewer: We note that your Data Availability Statement is currently as follows: All data generated or analysed during this study are included in this published article (and its Supplementary Information files).

Our answer: We provide the raw data as an Excel file, further we provide the R markdown file and the corresponding html output file. This output file contains a description of the data and the results for kappa and ICC.

When preparing to make the data available we re-run the analysis using a current version of R. The results for kappa and ICC were identical to previous analyses. We added 95% confidence intervals.

Therefore, SAS is no longer mentioned.

We updated the methods section accordingly.

Concern of the reviewer: We note you have included a table to which you do not refer in the text of your manuscript. Please ensure that you refer to Table 3 in your text; if accepted, production will need this reference to link the reader to the Table.

Our answer: Thank you very much for your feedback, we added the reference!

Reviewer 1:

the topic of the manuscript is interesting but some changes are necessary before taking it into consideration for publication. Here are my suggestions to imrpve it:

Concern of the reviewer: try to better describe the DPIA procedure to ensure the anonymity of patients. Explain what the acronym means and briefly what it consists of

Our answer: Thank you very much for your comment. We added additional information on the DPIA method in the „Materials and Methods“ section.

Concern of the reviewer: better describe about the analyzed 50 photographs. Please explain whether the photo is of a single tooth or a group of teeth. Which surface (mesial, vestibular, occlusal...), what type of photo (intra or extra-oral, occlusal, lateral, 3/4,). What camera was used? Did you use flash? Who took the photos? Where the patients have been enrolled?

Our answer: Thank you very much for your feedback. We added additional information to our „Materials and Methods“ section.

Concern of the reviewer: please explain wh etegories: low, medium, and high orthodontic experience. how these 3 groups have been created?

Our answer: Thank you for your comment, we have tried to make the information about the three groups more comprehensive.

Concern of the reviewer: Who the observers are: hygienists, students, dentists, orthodantists, dental assistents?

Our answer: Thank you very much for your comment, we have clarified the information about the evaluators in the „Materials and Methods“ section.

Concern of the reviewer: The observers were divided into 3 categories: low, medium, and high orthodontic experience. But how these 3 groups were created?

Our answer: Thank you for your comment, we have tried to make the information about the three groups more comprehensive in the „Materials and Methods“ section.

Concern of the reviewer: Statistics is well done.

Our answer: Thank you very much, we appreciate the positive feedback.

Concern of the reviewer: references: 47/59 are older than 10 years old.

Our answer: Thank you very much for your comment, we have used this opportunity to update our literature references; however, the important original sources have been retained.

Reviewer 2:

The authors find that the inter-rater reliability of three plaque indices varies based on the evaluators' orthodontic experience. The Attin and mBB indices show the highest agreement among evaluators, while the TQH index has the lowest. Orthodontic experience does not significantly affect reliability. Evaluators with little experience show the highest agreement with the Attin index. Calibration of raters is recommended to achieve more consistent classifications. The authors conclude that the Attin and mBB indices are preferable for patients with multibracket appliances, whereas the TQH index is less suitable. Training and calibration are essential for effective application of plaque indices.I would like to the authors for their valuable efforts on this study. It was a pleasure to read.

Our response: Thank you very much for the feedback, we truly appreciate it!

---

## [Editor Report · Decision Letter 1]

24 Mar 2025

Investigation of the inter-rater reliability of three different plaque indices used in patients with fixed orthodontic appliances

PONE-D-24-15938R1

Dear Dr. Erbe,

We’re pleased to inform you that your manuscript has been judged scientifically suitable for publication and will be formally accepted for publication once it meets all outstanding technical requirements.

Kind regards,

Prof. Dr. Sameh Attia, MS

Academic Editor

PLOS ONE

---

## [Editor Report · Acceptance letter]

PONE-D-24-15938R1

PLOS ONE

Dear Dr. Erbe,

I'm pleased to inform you that your manuscript has been deemed suitable for publication in PLOS ONE. Congratulations! Your manuscript is now being handed over to our production team.

Kind regards,

on behalf of

Dr. Sameh Attia

Academic Editor

PLOS ONE